# Predicting base editing outcomes with an attention-based deep learning algorithm trained on high-throughput target library screens

Kim F. Marquart [1,2,6], Ahmed Allam[3,6], Sharan Janjuha[2,6], Anna Sintsova[3,4], Lukas Villiger[2,5], Nina Frey[1,2], Michael Krauthammer[3✉] & Gerald Schwank [2✉]

Base editors are chimeric ribonucleoprotein complexes consisting of a DNA-targeting CRISPR-Cas module and a single-stranded DNA deaminase. They enable transition of C●G into T●A base pairs and vice versa on genomic DNA. While base editors have great potential as genome editing tools for basic research and gene therapy, their application has been hampered by a broad variation in editing efficiencies on different genomic loci. Here we perform an extensive analysis of adenine- and cytosine base editors on a library of 28,294 lentivirally integrated genetic sequences and establish BE-DICT, an attention-based deep learning algorithm capable of predicting base editing outcomes with high accuracy. BE-DICT is a versatile tool that in principle can be trained on any novel base editor variant, facilitating the application of base editing for research and therapy.

[1] Institute of Molecular Health Sciences, ETH Zurich, Zurich, Switzerland. [2] Department of Pharmacology and Toxicology, University of Zurich, Zurich, Switzerland. [3] Department of Quantitative Biomedicine, University of Zurich, Zurich, Switzerland. [4] Present address: Institute of Microbiology, ETH Zurich, Zurich, Switzerland. [5] Present address: McGovern Institute for Brain Research at MIT, Massachusetts Institute of Technology, Cambridge, MA, USA. [6] These authors contributed equally: Kim F. Marquart, Ahmed Allam, Sharan Janjuha. ✉email: michael.krauthammer@uzh.ch; schwank@pharma.uzh.ch

Base editors are CRISPR-Cas-guided single-strand DNA deaminases. They enable precise genome editing by directly converting a targeted base into another, without the requirement of DNA double-strand break formation and homology-directed repair from template DNA[1]. There are two major classes of base editors: cytosine base editors (CBEs) converting C•G into a T•A base pairs[2], and adenine base editors (ABEs) converting A•T into G•C base pairs[3]. The most commonly used base editors comprise a nickase (n) variant of SpCas9 to stimulate cellular mismatch repair and have either the rat cytosine deaminase APOBEC1 (CBE) or a laboratory-evolved *E. coli* adenine deaminase ecTadA (ABE) fused to their N-termini. Both base editor classes convert target bases in a ~5-nucleotide region within the protospacer target sequence (editing window), where the DNA strand that is not bound to the sgRNA becomes accessible to the deaminase[2,3].

A major limitation of base editors is their broad variation in editing efficiencies across different target sequences. These can be influenced by several parameters, including the consensus sequence preference of the deaminase[4], and the binding efficiency of the sgRNA to the protospacer[5]. While low editing rates on a target locus that contains a single C or A may be circumvented by extending the exposure time to the base editor, undesired 'bystander' editing of additional C or A bases in the editing window generally requires optimization by experimental testing of alternative base editor constructs. Potentially successful strategies are (i) exchanging the sgRNA to shift Cas9 binding upstream or downstream, (ii) using a base editor with a narrowed activity window[6], or (iii) using a base editor with a deaminase that displays a different sequence preference (e.g. activation-induced deaminase (AID) instead of rAPOBEC1 or ecTadA8e instead of ecTadA7.10)[7,8], However, experimental screening with the different available base editor and sgRNA combinations is laborious and time-consuming.

In this work, we develop a machine learning algorithm capable of predicting base editing outcomes of commonly used ABEs and CBEs on any given protospacer sequence in silico available via www.be-dict.org.

## Results

### Generation of large datasets for adenine- and cytidine base editing via high-throughput screening with self-targeting libraries. To capture base editing outcomes of SpCas9 CBEs and ABEs across thousands of sites in a single experiment, we generated a pooled lentiviral library of constructs encoding unique 20-nt sgRNA spacers paired with their corresponding target sequences (20-nt protospacer and a downstream NGG PAM site) (Fig. 1a). Our library included 23,123 randomly generated target sequences and 5,171 disease-associated human loci with transition mutations, comprising a comprehensive and diverse library for machine learning (Supplementary Data 1). Oligonucleotides containing the sgRNAs and corresponding target sequences were synthesized in a pool and cloned into a lentiviral backbone containing an upstream U6 promoter and a puromycin resistance cassette. HEK293T cells were then transduced at a 1000× coverage with a multiplicity of infection (MOI) of 0.5, and selected with puromycin. Next, cells were transfected with Tol2 compatible plasmids encoding for blasticidin resistance and one of the four commonly used base editors: ABEmax (containing ecTad7.10), CBE4max (containing rAPOPEC1), ABE8e (containing ecTadA-8e), and Target-AID (containing the AID ortholog PmCDA1) (Supplementary Fig. 1). Co-transfection with a Tol2 transposase plasmid allowed stable integration and prolonged expression of base editors. After 10 days in culture, cells were harvested, and genomic DNA was collected for

amplicon high-throughput sequencing (HTS) (Fig. 1b and see the 'Methods' section).

We observed high consistency between both experimental replicates (Pearson's $r^2 = 0.88$ (ABEmax), 0.86 (CBE4max), 0.92 (ABE8e), and 0.88 (Target-AID)) (Supplementary Fig. 2), indicating comprehensive and robust sampling of edited target sites. Mean base editing efficiencies (defined here as the fraction of mutant reads overall sampled reads of a target site) were 4.26% for ABEmax, 3.61% for CBE4max, 3.15% for ABE8e, and 3.13% for Target-AID (Supplementary Fig. 3). In line with previous studies, we observed maximum editing at position 6 (counting from PAM distal) with ABEmax, CBE4max, and ABE8e, and at position 3 for Target-AID (Fig. 1c–f)[7–10]. Interestingly, the editing window of ABE8e was broader than ABEmax, and that of Target-AID was shifted PAM-distally compared to CBE4max (Fig. 1e, f). Analysis of the trinucleotide sequence context, moreover, confirmed that ecTadA7.10 of ABEmax and rAPOBEC1 of CBE4max have a preference for editing at bases that are preceded by T (Fig. 1g, h)[10–13]. ecTadA7.10 additionally has an aversion for an upstream A and preference for a downstream C. Notably, ecTad-8e of ABE8e displayed a reduced sequence preference, although editing of bases that were preceded by an A was still largely disfavored (Fig. 1i). Compared to rAPOBEC1 PmCDA1 of Target-AID lacked the requirement of a preceding T for efficient editing, but motifs, where the targeted base is followed by a C, were disfavored (Fig. 1j).

### Development of BE-DICT, an attention-based deep learning model predicting base editing outcomes. Potentially predictive features that influence CRISPR/Cas9 sgRNA activity, such as the GC content and minimum Gibbs free energy of the sgRNA, did not influence base editing rates (Supplementary Fig. 4). This prompted us to utilize the comprehensive base editing data generated in the ABE and CBE target library screens for designing and training a machine learning model capable of predicting base editing outcomes at any given target site. We established BE-DICT (Base Editing preDICTion via attention-based deep learning), an attention-based deep learning algorithm that models and interprets dependencies of base editing on the protospacer target sequence. The model is based on multi-head self-attention inspired by the Transformer encoder architecture[14]. It takes a sequence of nucleotides of the protospacer as input and computes the probability of editing for each target nucleotide as output (Fig. 2a). The formal descriptions of the model and the different computations involved are reported in Supplementary Notes 1–3. In short, BE-DICT assigns a weight (attention-score) to each base within the protospacer (i.e. learned fixed-vector representation). The input mode is dichotomous, where bases with editing efficiencies above or equal mean editing were classified as edited, and bases below were classified as non-edited. The output is a probability score, reflecting the likelihood (between 0 and 1) with which a target base will be edited (C-to-T or A-to-G). To train and test the model, we included all target sequences with at least one classified base edit (8,558 for ABEmax; 9,534 for CBE4max; 3,416 for ABE8e; 10,177 for Target-AID). In order to reduce the tendency towards edited target sequences, which could result in an inherent bias of the prediction tool, we also added unedited target sequences at a ratio of 1:4 (Supplementary Data 1). For model training, we used ~80% of the dataset and performed stratified random splits for the rest of the sequences to generate an equal ratio (1:1) between the test and validation datasets. We repeated this process five times (denoted by runs), in which we trained and evaluated a model for every base editor separately for each run. BE-DICT performance was then plotted using the area under the receiver operating characteristic curve (AUC), and

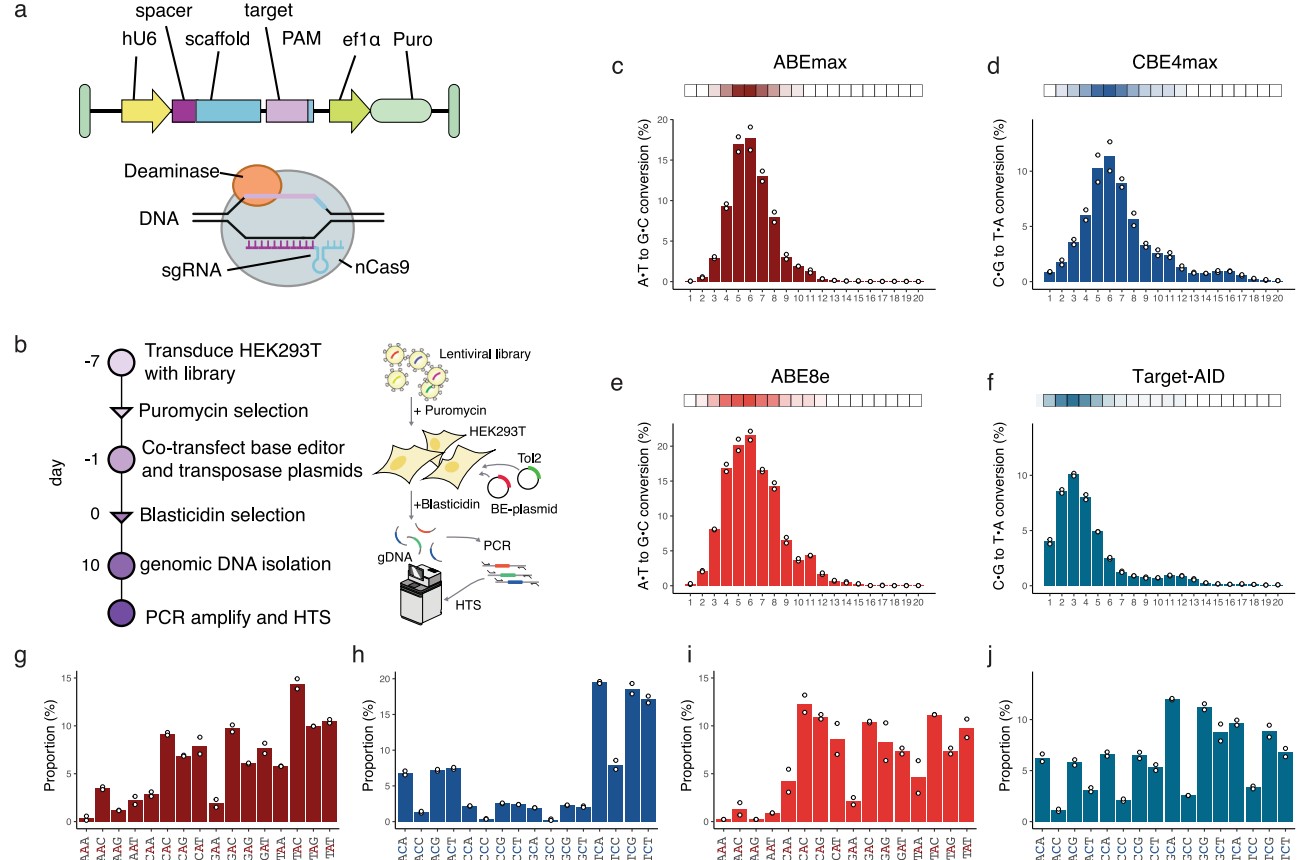

**Fig. 1 A high-throughput platform for assessing base editor activities. a** The design of the self-targeting library was adapted from refs. [27–30]. The lentiviral library contains the sgRNA expression cassette and the target locus on the same DNA molecule. The sgRNA (spacer and scaffold) is transcribed under the control of a U6 promoter and is designed to direct the base editor (nCas9-deaminase fusion) to the 20-nt sequence upstream of the protospacer adjacent motif (PAM). hU6 human U6 promoter, ef1α elongation factor 1α promoter, nCas9 nickase Cas9, sgRNA single-guide RNA, Puro puromycin selection marker. **b** Overview of library screening. **c–f** Base editor profiles for loci above mean editing efficiency for **c** ABEmax, **d** CBE4max, **e** ABE8e, and **f** Target-AID. The plot shows the average efficiency of A-to-G or C-to-T base conversions at each position across the protospacer target sequence. The top horizontal bar illustrates the favored activity window of the respective deaminase. **g–j** Proportion of the different tri-nucleotide motifs for loci above mean editing efficiency for **g** ABEmax, **h** CBE4max, **i** ABE8e, and **j** Target-AID. The number of analyzed target sequences shown in **a–j** are as follows: $n = 8,558$ (ABEmax); 9,534 (CBE4max); 3,416 (ABE8e); and 10,177 (Target-AID).

the area under the precision-recall curve (AUPR). For all four models, an AUC of between 0.92–0.95 and AUPR between 0.733–0.806 was achieved (Fig. 2b–e). Notably, at positions within the activity window where we have a balanced distribution of edited vs. unedited substitute bases, BE-DICT performed with significantly higher accuracy than a per position majority class predictor—a baseline model that predicts nucleotides conversions as a Bernoulli trial, using maximum-likelihood estimation for computing the probability of editing success at each position (Fig. 2f–i, Supplementary Fig. 5).

**BE-DICT can be utilized to predict editing efficiencies at endogenous loci and predominantly puts attention to bases flanking the target base.** Base editing at endogenous loci may also be affected by protospacer sequence-independent factors, such as chromatin accessibility. We, therefore, tested the accuracy of BE-DICT in predicting base editing outcomes at 18 separate endogenous genomic loci for ABEmax and ABE8e, and 16 endogenous genomic loci for CBE4max and Target-AID. HEK293T cells were co-transfected with plasmids expressing the sgRNA and base editor, and genomic DNA was isolated after 4 days for targeted amplicon HTS analysis. Across all tested loci we observed a strong correlation between experimental editing rates and the BE-DICT probability score (Pearson's $r = 0.78$ for ABEmax, 0.68 for

CBE4max, 0.57 for ABE8e, and 0.64 for Target-AID; Fig. 3a–d; Supplementary Data 2). Further validating our model, BE-DICT also accurately predicted base editing efficiencies from previously published experiments (Pearson $r = 0.82$ for ABEmax, 0.71 for CBE4max, 0.91 for ABE8e, and 0.76 for Target-AID; Supplementary Fig. 7; Source Data)[8,15]. These results demonstrate that the BE-DICT probability score can be used as a proxy to predict ABEmax and CBE4max editing efficiencies with high accuracy.

The attention-based BE-DICT model provides insights (attention scores) for each position within the protospacer with regard to the position's influence on the editing outcome. These attention scores provide a proxy for identifying relevant motifs and sequence contexts for editing outcomes. Interestingly, we found that for all base editors (ABEmax, CBE4max, ABE8e, and Target-AID) BE-DICT attention was mainly focused on bases flanking the target base and on the target base position itself (Fig. 4a–d). In addition, we observed that base attention patterns were dependent on the position of the target base, and occasionally consisted of complex gapped motifs rather than consecutive bases (Supplementary Fig. 6) underscoring the necessity of using machine learning for predicting base editing outcomes.

**Development of the BE-DICT bystander module.** Multiple A or C nucleotides within the editing window can lead to bystander

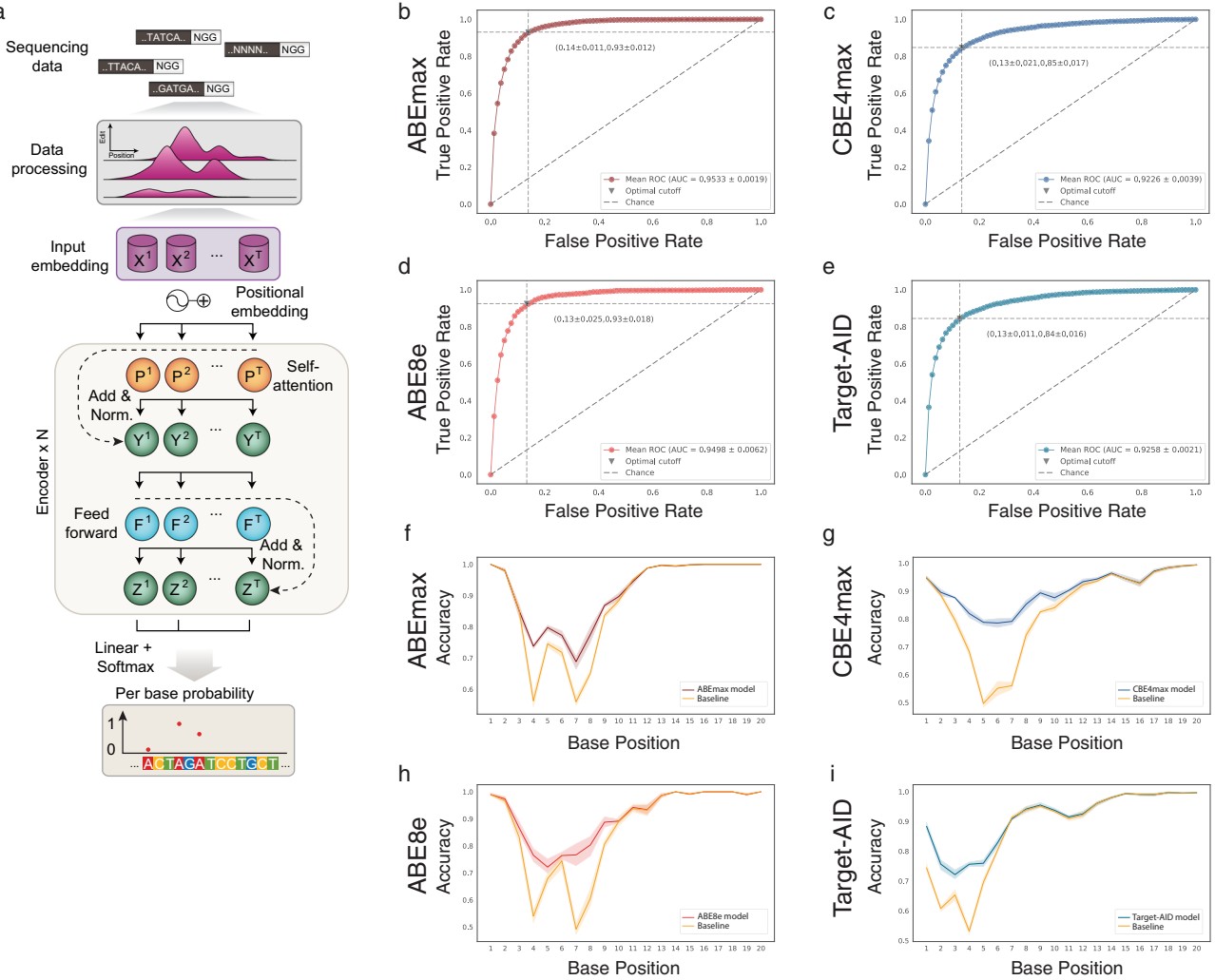

**Fig. 2 BE-DICT: A machine learning model for predicting base editing outcomes. a** Design of an attention-based deep learning algorithm to predict base editing probabilities. Given a target sequence, the model returns a confidence score to predict the chance of target base conversions. The model has three main blocks: (1) An embedding block that embeds both the nucleotide and its corresponding position from one-hot encoded representation to a dense fixed-length vector representation. (2) An encoder block that contains a self-attention layer (with multi-head support), layer normalization[31] and residual connections, and a feed-forward network. (3) An output block that contains a position attention layer, and a classifier layer. **b–e** The average AUC achieved across five runs (interpolated) for models trained on data from high-throughput base editing experiments. **f–i** Line plot of per-position accuracy of the trained models across five individual runs for base editors in comparison to the accuracy of majority class baseline predictor. Standard deviation is depicted as a band along with the line plot.

base conversions. These are often undesired, in particular, if they induce coding mutations in the targeted gene. Given that BE-DICT per-base models the 'marginal probability' of target base editing by providing a probability score whether a single base will be edited, it does not directly predict the editing efficiency of a locus (i.e. it cannot predict co-occurrences of target base- and bystander editing). Therefore, we next developed an extension module of BE-DICT, which is adapted to predict the relative proportions of all different editing outcomes (combinations of target base and bystander transitions) per target locus (BE-DICT bystander module—Fig. 5a). The model is based on an encoder–decoder architecture (adapting the Transformer architecture used in the BE-DICT per-base model), which takes a sequence of nucleotides of the protospacer as input, and computes the probability of the different output sequences (i.e. probabilities for all combinations of sequences with target-based and bystander transitions, as well as the probability of observing a wild-type sequence) (Fig. 5a). The formal description of the model is reported in Supplementary Notes 2 and 3. In short, it

uses an encoder module that computes a vector representation for each nucleotide in the input protospacer sequence, and then uses a decoder that has the same components of the encoder module with the exception of a masked self-attention and cross-attention layer. The masked self-attention layer acts as an "autoregressive layer", ensuring the use of only past information while computing the probability of the output. The cross-attention layer learns what parts of the input sequence are important when computing the vector representation of the nucleotides in the output sequence, subsequently allowing the model to compute the probability of each output sequence. For model training, we used the edited input sequences from the ABEmax-, CBE4max-, ABE8e-, and Target-AID library screens that were already used to train and test the BE-DICT per-base model, and again partitioned them in an 8:1:1 ratio for training, testing, and validation. Unlike for the per-base BE-DICT model, however, the outcome is non-binary and represented the frequencies of all outcomes on the target sites (unedited read and the different edited outcomes) for a given input sequence (i.e. protospacer). The trained BE-DICT

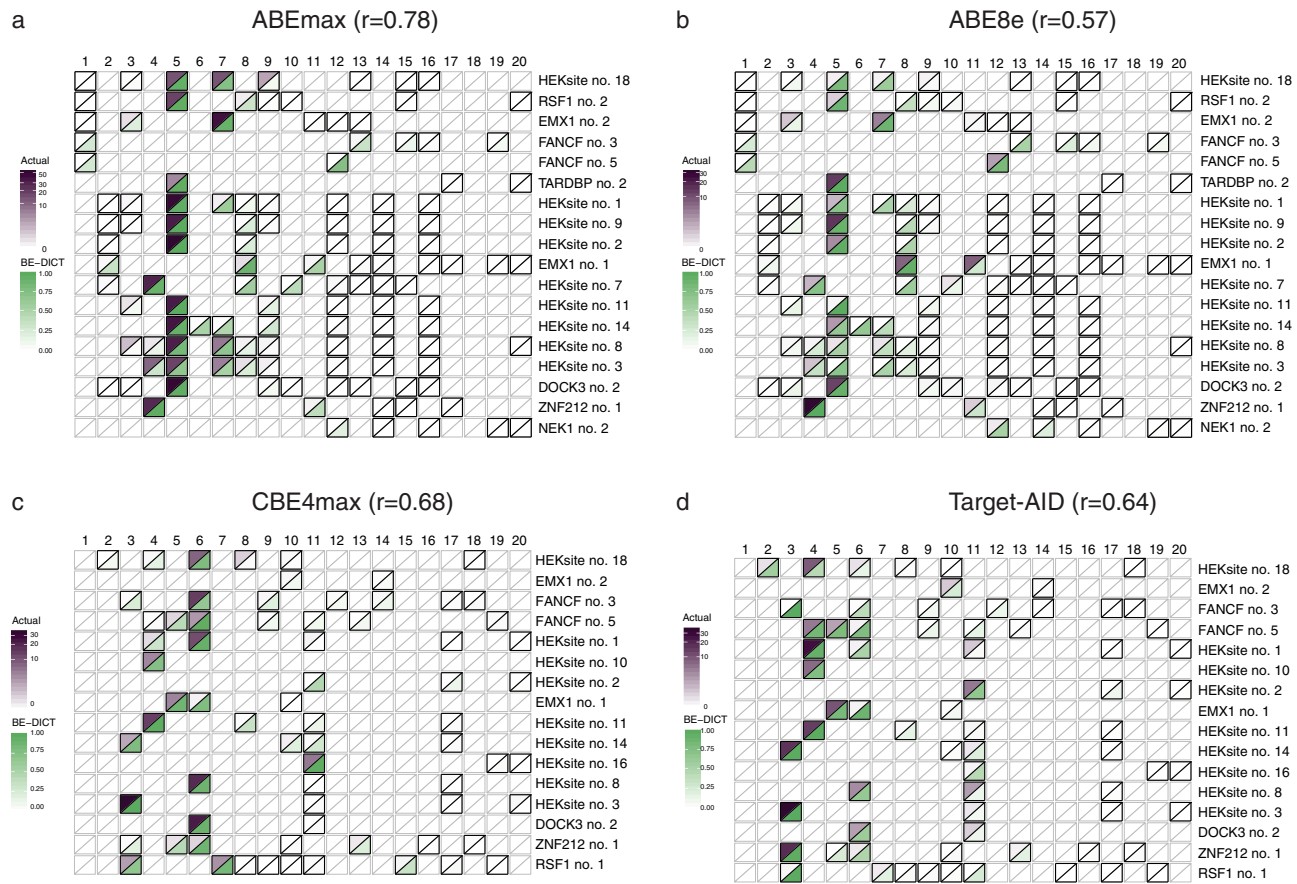

**Fig. 3 BE-DICT accurately predicts base editing activities on endogenous genomic loci in HEK293T cells.** Endogenous genomic target sequences with at least two substrate nucleotides were targeted separately by co-transfection of the sgRNA and base editors **a** ABEmax, **b** CBE4max, **c** ABE8e, and **d** Target-AID. Heatmap shows the BE-DICT prediction score (green) and experimentally observed target base conversion (purple). Substrate bases for the respective base editor are outlined in bold. Pearson's correlation (*r*) for all target bases is shown.

bystander module predicted various possible editing outcomes per target sequence, including combinations with multiple base conversions (Fig. 5b, c). Importantly, the performance was reliable for all four base editors, as we achieved strong correlations between predicted and experimentally observed sequence proportions in the validation datasets (Pearson's *r* = 0.86 for ABEmax, 0.94 for CBE4max, 0.66 for ABE8e, and 0.97 for Target-AID; Supplementary Fig. 8).

Recently, two other machine learning models capable of predicting base editing outcomes have been developed. BE-Hive[10], which is a deep conditional autoregressive model, and DeepBaseEditor[13], which is based on a two-hidden layer convolutional neural network framework. Contrary to the BE-DICT bystander module that directly predicts the proportions of all outcomes at the target locus, both models separately predict the proportions of edited outcomes and the overall editing efficiency of the target site, and combining both values is required to estimate the frequency of precise target base conversion without bystander mutations (Fig. 5d). Since also BE-Hive and DeepBaseEditor have been trained and applied on TadA7.10-ABE and APOBEC1-CBE datasets, we decided to compare their performance to our attention-based machine learning model. First, we only benchmarked the ability of the three models to predict the proportions of edited outcomes. Therefore, we adapted the BE-DICT bystander model to only calculate the proportions of edited outcomes, comparable to the BE-Hive bystander and DeepBaseEditor proportion models. When applied to the high-throughput datasets of the three studies, all models

achieved similarly good correlations with the experimentally observed values using Pearson's correlation (Fig. 5e, f; Supplementary Fig. 10a, b) or Spearman's correlation (Supplementary Fig. 9). Next, we compared the ability of the three models to predict the proportions of all outcomes (including the wildtype sequence) at a target locus. Again, predicted values correlated well with the experimentally observed values for all three models (Fig. 5g, h; Supplementary Fig. 10c, d). Interestingly, the performance of the three models was not substantially affected by the differences in the experimental setup of the three datasets (Fig. 5e–h; Supplementary Fig. 10), suggesting that they can tolerate variations in experimental procedures between laboratories. Confirming this hypothesis, when BE-DICT was retrained on the ABE datasets of Song et al. [13] (HT_ABE_Train), correlations between predicted and experimentally observed editing outcomes on the HT_ABE_Test dataset of Song et al. increased only incrementally to *r* = 0.94 (Supplementary Fig. 11). Altogether, we conclude that the three machine learning models operate robustly on different experimental datasets and with comparable accuracy.

## Discussion

In this study, we used a high-throughput approach to assess the activity and accuracy of base editors on thousands of target sites. The resulting datasets were used to train BE-DICT, a deep learning model capable of accurately predicting the editing of a target nucleotide and surrounding bystander nucleotides.

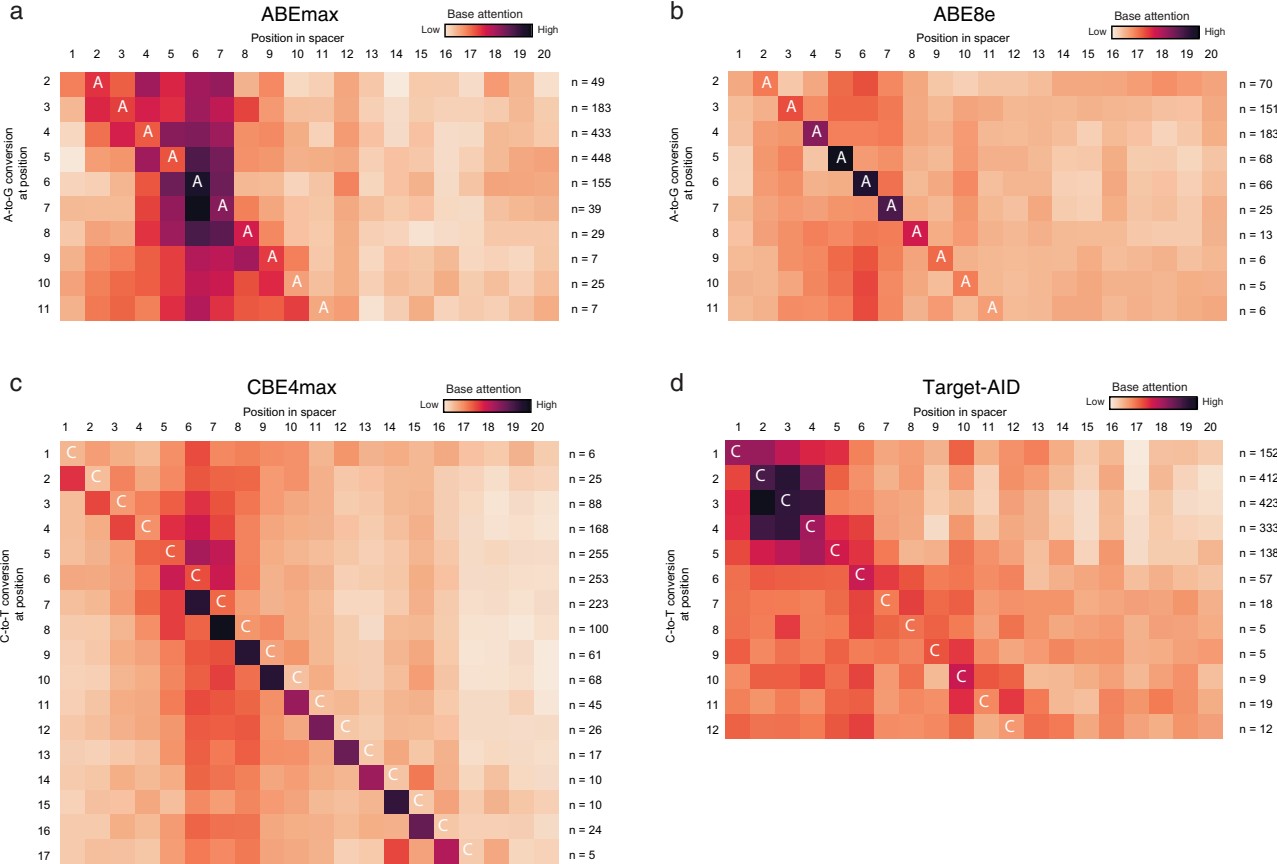

**Fig. 4 BE-DICT attention patterns for edited target bases.** Attention weight is indicated by color from light to dark. Mean-aggregate attention scores of target sequences (BE-DICT test dataset) predicted to be edited at the respective position for **a** ABEmax, **b** CBE4max, **c** ABE8e, and **d** Target-AID are shown. The number of sequences used for the analysis of the respective target base position is indicated as *n*.

Currently, BE-DICT is trained on datasets for ABEmax, CBE4-max, ABE8e, and Target-AID, and the respective models are freely accessible at www.be-dict.org. As the algorithm is versatile, it could also be adopted to various other base editor variants in the future, enabling researchers a priori selection of an optimal base editor for their target locus. This is essential when base editors are applied in genome editing therapies, where a disease-causing point mutation should be repaired without inducing bystander edits[1,10]. In addition, it is also important for the application of BEs in genetic screening, as bystander mutations might affect the phenotype that is caused by the target base conversion[16].

Recently, two other machine learning models have been developed that also predict the proportions of base editing outcomes (BE-Hive and DeepBaseEditor). We extensively compared the three models and found that they perform with similar accuracy. Notably, in addition to the bystander module, BE-DICT also offers a per-base module. While this module only gives the probability with which a target base is edited (or not), and also cannot predict combinations of target base and bystander conversions, it offers the feature of identifying motifs that are highly favored or disfavored by currently available base editors. This could potentially guide researchers to develop novel base editor variants with improved activities in the future. In addition, the BE-DICT per-base module implicitly models the marginal editing probability at each position. Thus, unlike the other models (i.e. bystander models), for which complexity of the search space increases exponentially with the nucleotide number, the BE-DICT per-base module exhibits a quadratic complexity, which may be further improved by scaling the self-attention layer to O(n)

complexity[17]—in principle enabling the model to consider a much wider sequence context far beyond the protospacer target site. Overall, the BE-DICT modules can accurately predict base editing outcomes and can guide researchers in designing base editing experiments.

## Methods

**Oligonucleotide-library design.** The custom oligonucleotide pool containing pairs of sgRNA and corresponding target sequences was purchased from Twist Bioscience. The library includes 23,123 random DNA sequences and 5,171 disease loci theoretically targetable using base editors. Designed oligonucleotides include the following elements: The G/20N spacer and SpCas9 gRNA scaffold, a six-nucleotide randomized barcode, the corresponding target locus containing the PAM, and a second six-nucleotide randomized barcode (Supplementary Note 4). Randomized DNA sequences of 20 bp length and 1:1:1:1 proportion of each nucleotide were generated using a custom Python script to form a random sequence library. The disease loci were selected from the NCBI ClinVar[18] database (accessed in May 2019) using the following criteria: (a) all disease-associated SNPs were accessed and restricted to pathogenic and monogenic filters (b) SNPs were further restricted to the possible base conversions targetable by ABEs (A-to-G) and CBEs (C-to-G). (c) Genomic region flanking the SNP genomic coordinates were extracted from UCSC server (http://genome.ucsc.edu/). (d) The sequences were then scanned presence of an NGG PAM 8–18 bases away from the target base. Only SNPs passing these filtering criteria were included in the study and were then appended to the list of aforementioned random sequences to form the final library.

**Plasmid-library preparation.** The plasmid library containing the sgRNA and the corresponding target sequence was prepared using a one-step cloning process to prevent uncoupling of the sgRNA- and target sequence. The oligonucleotide pool was PCR-amplified in 10 cycles (Primers stated in Supplemental Information) and KAPA® HiFi HotStart Polymerase (Roche) following the manufacturer's instructions. The resulting amplicons were purified using 0.8 × volumes of para-magnetic AMPure XP beads (Beckman Coulter) following the manufacturer's instructions for PCR cleanup. We digested the Lenti-gRNA-Puro plasmid with

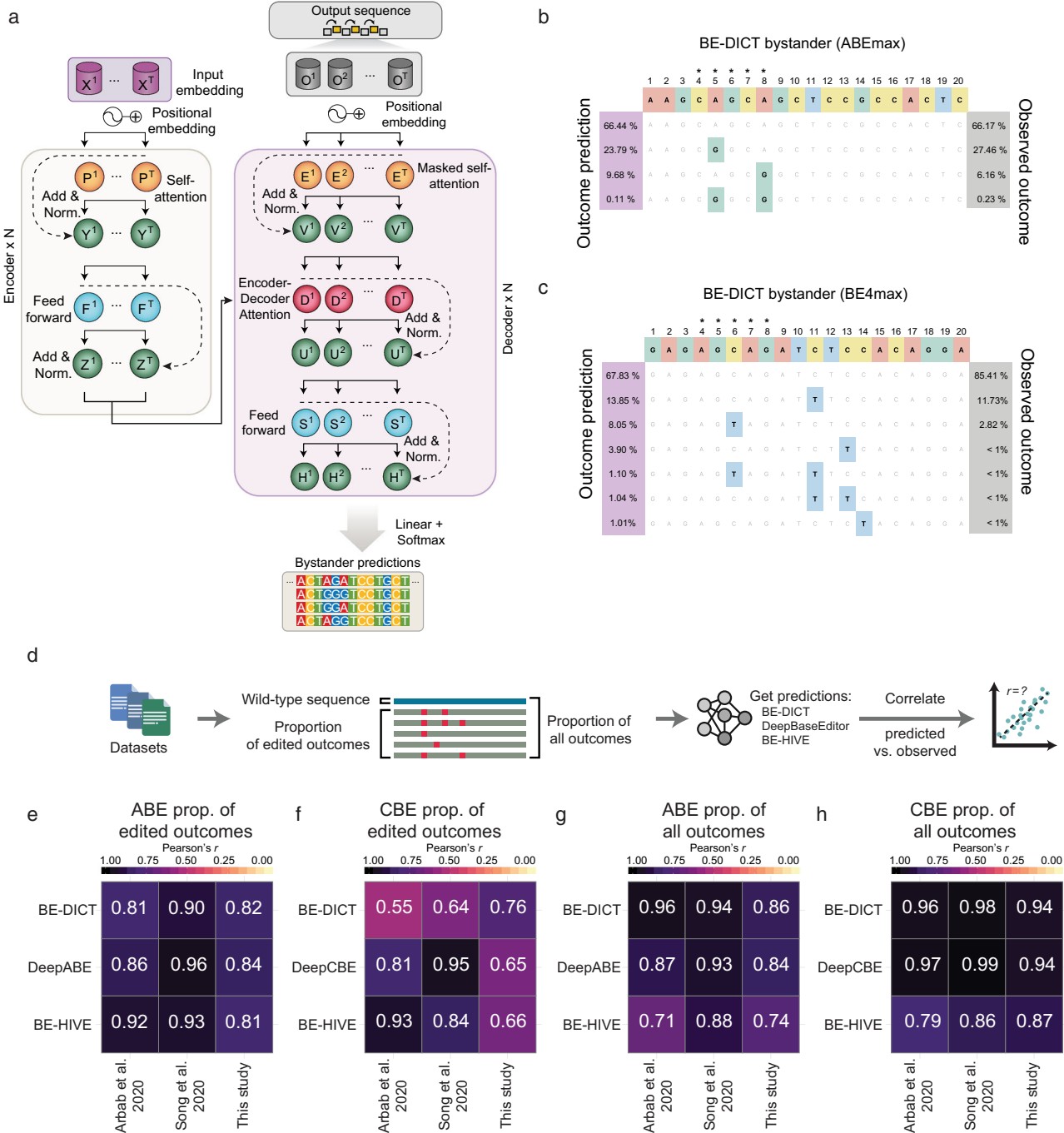

**Fig. 5 BE-DICT bystander model. a** An extension of the BE-DICT algorithm enables the prediction of the frequency of reads with bystander mutations. **b, c** Graphical representations of predicted and experimentally observed allele frequencies for an ABEmax and CBE4max target site. **d** Workflow for benchmarking BE-DICT with BE-HIVE and DeepBaseEditor. **e–h** Performance evaluation of different machine learning models for ABE and CBE on prediction of the proportion of edited outcomes (**e, f**) and the prediction of all outcomes (**g, h**). Pearson's correlation (*r*) was calculated by comparison of predicted versus measured base editing outcome proportions in datasets published by Arbab et al. and Song et al. [10, 13], and in this study. The number of analyzed outcomes in the dataset from Arbab et al. are $n = 7,743$ (ABE edited outcomes), $n = 7,537$ (CBE edited outcomes), $n = 9,008$ (ABE all outcomes), and $n = 8,895$ (CBE all outcomes) arising from a total of 1,265 unique sequences for ABE and 1,358 sequences for CBE. The number of analyzed outcomes in the dataset from Song et al. is $n = 1,767$ (ABE edited outcomes), $n = 2,332$ (CBE edited outcomes), $n = 2,204$ (ABE all outcomes), and $n = 2,807$ (CBE all outcomes) arising from a total of 437 unique sequences for ABE and 475 sequences for CBE. The number of analyzed outcomes in the dataset from this study is $n = 3,844$ (ABE edited outcomes), $n = 4,502$ (CBE edited outcomes), $n = 5,510$ (ABE all outcomes), and $n = 6,176$ (CBE all outcomes) arising from a total of 1,667 unique sequences for ABE and 1,675 sequences for CBE.

BsmBI restriction enzyme (New England Biolabs, NEB) for 12 h at 55 °C. Lenti-gRNA-Puro was a gift from Hyongbum Kim (Addgene no. 84752[19]). After digestion, the plasmid was treated with calf intestinal alkaline phosphatase (NEB) for 30 min at 37 °C and gel purified with a NucleoSpin Gel and PCR Clean-up Mini kit (Macherey-Nagel). The oligo-pool amplicons were assembled into the linearized

Lenti-gRNA-Puro plasmid using NEBuilder HiFi DNA Assembly Master Mix (NEB) for 1 h at 50 °C. The product was precipitated by adding one volume of Isopropanol (99%), 0.01 volumes of GlycoBlue coprecipitant (Invitrogen), and 0.02 volumes of 5 M NaCl solution. The mix was vortexed for 10 sec and incubated at room temperature for 15 min followed by 15 min of centrifugation ($15,000 \times g$).

The supernatant was discarded and replaced by two volumes of ice-cold ethanol (80%). Ethanol was removed immediately and the pellet was air-dried for 1 min. The pellet was dissolved in TE buffer (10 mM Tris, 0.1 mM EDTA) and incubated at 55 °C for 10 min. In 12 transformation replicates, 2 μL of plasmid library were transformed per 50 μL of electrocompetent cells (ElectroMAX Stabl4, Invitrogen) using a Gene Pulser II device (Bio-Rad). Transformed cells were recovered in S.O.C. media (from ElectroMAX Stabl4 kit) for 1 h and spread on Luria–Bertani agar plates (245 × 245 mm, Thermo Fisher Scientific) containing 100 μg/mL ampicillin. After incubation at 30 °C for 14 h, the colonies were scraped and plasmids were purified using a Plasmid Maxiprep kit (Qiagen).

**Cell culture**. HEK293T (ATCC CRL-3216) were maintained in DMEM plus GlutaMax (Thermo Fisher Scientific), supplemented with 10% (vol/vol) fetal bovine serum (FBS, Sigma-Aldrich) and 1 × penicillin–streptomycin (Thermo Fisher Scientific) at 37 °C and 5% CO$_2$. Cells were maintained at confluency below 90% and passaged every 2–3 days. Cells were discarded after 15 consecutive passages.

**Packaging of guide RNA library into lentivirus**. Transfection in T225 cell culture flasks was conducted as follows: 3.4 μg pCMV-VSV-G (lentiviral helper plasmid; addgene plasmid no. 8454, a gift from B. Weinberg[20]), 6.8 μg psPAX2 (lentiviral helper plasmid; addgene plasmid no. 12260, a gift from D. Trono) and 13.6 μg target library plasmid were mixed in 651 μL serum-free Opti-MEM (Thermo Fisher Scientific), supplied with 195 μL of polyethyleneimine (PEI, 1 mg/mL), vortexed for 10 s and incubated for 10 min. 25 mL of DMEM was added to the transfection mix and gently pipetted to the cells at ~70% confluency. The medium was changed 1 day after transfection. 2 days later, supernatant-containing lentiviral particles were harvested and filtered using a Filtropur S 0.4 (Sarstedt) filter. The virus suspension was ultracentrifuged (20,000×g) for 2 h. Aliquots were frozen at −80 °C until use.

**Cloning of base editors**. Expression plasmids were constructed using isothermal assembly (NEBuilder® HiFi DNA Assembly Cloning Kit, NEB) or restriction digest and ligation using T4 ligase (NEB). Plasmid p2T-ABE8e-BlastR was generated by ligation of the ABE8e transgene (AgeI-NotI digest of pCMV-ABE8e) into the Tol2 compatible backbone (AgeI-NotI-EcoRV digest of p2T-CMV-ABEmax-BlastR). Plasmid p2T-Target-AID-GFP-BlastR was generated by isothermal assembly of PCR amplified Target-AID transgene from the template plasmid pRZ762 and the Tol2 compatible backbone (AgeI-NotI-EcoRV digest of p2T-CMV-ABEmax-BlastR). PCR was conducted using NEBNext® High-Fidelity 2X PCR Master Mix (NEB). Plasmids p2T-CMV-ABEmax-BlastR (Addgene no. 152989) and ABE8e (Addgene no. 138489) were gifts from David Liu[8,10]. Target-AID (pRZ762, Addgene plasmid no. 131300) was a gift from Keith Joung[21].

**Pooled base editor screens**. T175 cell culture flasks were seeded with HEK293T cells and cultured to reach 70–80% confluence. 10 μg/mL polybrene was added to the media and the gRNA-pool lentivirus was transduced at a MOI of 0.5 and a calculated coverage of 1,000 cells per gRNA. One day after transduction, cells were supplied with fresh media with 2.5 μg/mL puromycin. After 9 days of puromycin selection, the respective base editor plasmids (9.25 μg) and helper plasmid (9.25 μg of pCMV-Tol2) were transfected in a 1:3 DNA:PEI ratio per T175 flask. 7.0 × 10$^7$ cells were transfected for each replicate independently. One day after transfection, cells were supplied with fresh media with 7.5 μg/mL blasticidin, depleting <50% of the initially transfected cells during selection. Ten days later, cells were detached and genomic DNA was extracted using a Blood & Cell Culture DNA Maxi kit (Qiagen) according to the manufacturer's instructions. Base editor plasmid p2T-CMV-BE4max-BlastR (Addgene no. 152991) was a gift from David Liu[10]. Helper plasmid pCMV-Tol2 (Addgene plasmid no. 31823) was a gift from Stephen Ekker[22].

**Guide RNA cloning**. The vector backbone (lentiGuide-Puro, Plasmid no. 52963, a gift from F. Zhang[23]) was digested with Esp3I (NEB) and treated with rSAP (NEB) at 37 °C for 3 h and gel-purified on a 0.5% agarose gel. For sgRNA phospho-annealing (sequences in Supplementary Note 4), 1 μL of sgRNA top and bottom strand oligonucleotide (100 μM each), 1 μL 10× T4 DNA Ligase Buffer, 1 μL T4 PNK (NEB) and 6 μL H$_2$O were mixed and incubated in a thermocycler (BioRad) using the following program: 37 °C for 30 min, 95 °C for 5 min, ramp down to 25 °C at a rate of 5 °C/min. The annealed oligonucleotides were diluted 1:100 in H$_2$O and ligated into the vector backbone using 50 ng digested lentiGuide-Puro plasmid, 1 μL annealed oligonucleotide, 1 μL 10× Ligase Buffer (NEB), and 1 μL T4 DNA Ligase (NEB) in a 10 μL reaction (filled up to total volume with H$_2$O). The ligation mix was incubated at room temperature for 3 h and transformed into NEB Stable Competent *E. coli* (C3040H) following the manufacturer's instructions. Correct assembly of the sgRNA into the backbone was confirmed by SANGER-Sequencing (Microsynth) and plasmids were isolated using a GeneJET Plasmid Miniprep Kit (Thermo Fisher Scientific) following the manufacturer's instructions.

**Arrayed sgRNA transfections**. For base editor DNA on-target experiments HEK293T cells were seeded into 96-well flat-bottom cell culture plates (Corning), transfected 24 h after seeding with 150 ng of the base editor, and 50 ng of gRNA expression plasmid and 0.5 μL of Lipofectamine 2000 (Invitrogen) per well. One day later, the medium was removed and cells were detached using one drop of TrypLE (Gibco) per well, resuspended in a fresh medium containing 2.5 μg/μL puromycin, and plated again into 96-well flat-bottom cell culture plates. Cells were detached 4 days after transfection and pelleted by centrifugation. To obtain genomic DNA, cells were resuspended in 30 μL 1× PBS and 10 μL of lysis buffer (4× Lysis Buffer: 10 mM Tris–HCl at pH 8, 2% Triton X, 1 mM EDTA, and 1% freshly added Proteinase K (Qiagen)) was added to the cell suspension. The lysis was performed in a thermocycler (Bio-Rad) using the following program: 60 °C, 60 min; 95 °C, 10 min; 4 °C, hold. The lysate was diluted to a final volume of 100 μL using nuclease-free water and 1 μL of each lysate was used for the subsequent PCR. Plasmids pCMV-ABEmax-P2A-GFP (Addgene plasmid no. 112101), pCMV-BE4max-P2A-GFP (Addgene plasmid no. 112099), and pCMV-ABE8e (Addgene plasmid no. 138489) were gifts from David Liu[9].

**Library preparation for targeted amplicon sequencing of DNA**. Next-generation sequencing (NGS) preparation of DNA was performed as previously described[24]. In short, the first PCR was performed to amplify genomic sites of interest with primers containing Illumina forward and reverse adaptor sequences (see Supplementary Note 4 for oligonucleotides used in this study). To cope with high DNA input used for pooled screens, the Herculase II Fusion DNA Polymerase (Agilent) was used according to the manufacturer's instructions. For all other NGS-PCRs on genomic DNA and the second NGS-PCR, the NEBNext High-Fidelity 2 × PCR Master Mix (NEB) was used according to the manufacturer's instructions. In brief, 0.96 mg of genomic DNA per replicate of the pooled gRNA screen was amplified in 24 cycles for the first PCR using 10 μg gDNA input in 100 μl reactions. For arrayed gRNA experiments, 1 μL of the cell lysate per replicate was used in a 12.5 μL PCR reaction. The first PCR products were cleaned with paramagnetic beads, then the second PCR was performed to add barcodes with primers containing unique sets of p5/p7 Illumina barcodes (analogous to TruSeq indexes). The second PCR product was again cleaned with paramagnetic beads. The final pool was quantified on the Qubit 4 (Invitrogen) instrument. Pooled sgRNA screens were sequenced single-end on the Illumina NovaSeq 6000 machine using an S1 Reagent Kit (100 cycles). Arrayed gRNA experiments were sequenced paired-end (2 × 150) on the Illumina MiSeq machine using a MiSeq Reagent Kit v2 Nano.

**HTS analysis**. Fastq reads obtained from deep sequencing were trimmed up to the guide sequence and the flanking barcodes (6 + 20 + 3 + 6 = 35 bp) by removing the Illumina adapters and the plasmid scaffold sequences using Cutadapt v2.2[25]. The trimmed reads were then mapped using bowtie2 v2.3.5.1[26] with default parameters to a reference consisting of target sequences making up the library and the proportions of edited reads carrying distinct base conversions were tabulated. The loci were further filtered for reading depth of at least 100 and above mean editing efficiency for each base editor library. Edited reads were further restricted to C-to-T (in case of CBE) or A-to-G (in case of ABE) conversions in the protospacer. Only reads passing the filtering criteria were used for further analysis.

**Statistics and reproducibility**. All statistics were performed using R 3.5.2 and Python3. The editing efficiency for each gRNA was calculated according to the following formula:

$$Editing\ percentage = \frac{Read\ count\ of\ edited\ base}{Total\ read\ count\ of\ the\ target\ sequence} \times 100$$

$$Overall\ editing\ percentage = \frac{Sum\ of\ read\ count\ of\ all\ edited\ reads\ for\ the\ target}{Total\ read\ count\ of\ the\ target\ sequence} \times 100$$

Base editing experiments were performed at least in independent biological duplicates. Unless indicated otherwise, BE-DICT model predictions were collected from five runs.

DeltaG was calculated using the online resource at http://www.unafold.org/Dinamelt/applications/quickfold.php.

**Benchmarking three machine learning models**. To benchmark the ability of the three models to predict the proportions of edited outcomes (with and without the wild-type sequences), we used published data from Song et al. (HT_ABE_Train, HT_ABE_Test, HT_CBE_Train, HT_CBE_Test)[13], Arbab et al. (HEK293T_12k-Char_ABE, HEK293T_12kChar_BE4)[10] and our data for both ABEmax and CBE4max editors. We prepared the data for each model (Supplementary Data 3), such that each model has to predict editing combinations (i.e. canonical transitions A-to-G and C-to-T) anywhere in the protospacer sequences (i.e. 20 nucleotides). Each model predicted the proportion of edit outcomes for each of the three datasets and the results were correlated to the "ground-truth" observed edited outcomes in the respective data. Similarly, we compared the ability of the three models to predict the proportions of all outcomes (including the wildtype sequence) occurring in the protospacer sequence for each dataset. For the Song et al. [13] models (i.e. DeepBaseEditor) we used the models provided in the reported web application and

parsed the output to extract predicted values for the corresponding protospacers reported in Supplementary Data 3. Similarly, for Arbab et al. [10] (i.e. BE-Hive), we used the models reported in the code repository (on Github). For both models (DeepBaseEditor and BE-hive), we used "bystander" and "overall efficiency" models (as described in the respective manuscripts and online web tools) to predict the proportions of all outcomes (including the wildtype sequence) in the protospacers corresponding to the three datasets used.

**Reporting summary**. Further information on research design is available in the Nature Research Reporting Summary linked to this article.

## Data availability
We have provided the data sets used in this study as Supplementary Data 1–3. Additional information on the BE-DICT algorithm and DNA used in this study can be found in the Supplementary Information (Supplementary Notes 1–4). DNA-sequencing data are deposited under accession number PRJNA735610 (NCBI Sequence Read Archive) (https://www.ncbi.nlm.nih.gov/bioproject/?term = PRJNA735610). Source data are provided with this paper.

## Code availability
We have made the source code for BE-DICT and the custom Python scripts used to train and evaluate the models available on GitHub at https://github.com/uzh-dqbm-cmi/crispr and https://github.com/sharan-j/GenCountTable. The web application for BE-DICT, which predicts the base editing patterns in the DNA sequence is available at www.be-dict.org.

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

## Acknowledgements
We thank the Functional Genomics Center Zurich (FGCZ) for their help and support in next generation sequencing. This work was supported by the SNF (310030_185293), the URPP 'Human Reproduction Reloaded', and the Helmut Horten Foundation. S.J. holds an EMBO Long-Term Fellowship (ALTF 873-2019), and K.F.M. holds a PHRT iDoc Fellowship (PHRT_324).

## Author contributions
K.F.M. designed the study, performed experiments, and analyzed data. A.A. and A.S. designed and generated the machine learning algorithm BE-DICT and analyzed data. S.J. designed the study and analyzed the high-throughput sequencing data. L.V. and N.F. performed experiments. M.K. and G.S. designed and supervised the research and wrote the manuscript. K.F.M., A.A., S.J. and A.S. edited the manuscript. All authors approved the final version of the manuscript.

## Competing interests
The authors declare no competing interests.
