## [Peer Review File · Nature Communications]

Reviewers' Comments:

Reviewer #1:

Remarks to the Author:

The authors generated large datasets of CBE and ABE outcomes using lentivirally integrated target sequences and then developed BE-DICT, an attention-based deep learning algorithm capable of predicting base editing outcomes. The following issues should be addressed.

Major comments

1. The novelty of this study has been reduced by two recent studies (Arbab et al. Cell 2020; Song et al. Nat. Biotechnol. 2020). Furthermore, unlike the current study, these published studies considered combinations of base conversions. Given that the majority of users of BE-dict would want to know about such combinations, BE-dict should provide this information as well to reach at least a similar level as these previous studies.
2. The overall editing efficiencies are fairly low for both ABE_{max} and CBE_{4max} (median overall editing efficiency ABE_{max}= 2.2%, CBE_{4max}= 2.4%). This finding might be attributable to low transfection efficiencies, given that the authors neither checked the transfection efficiencies nor enriched transfected cells. These low editing efficiencies imply that the large dataset used in this study is of poor quality.
3. Fig. 1E, F; these results should be compared with those of the two recent studies (Arbab et al. Cell 2020; Song et al. Nat. Biotechnol. 2020).
4. The authors used an attention algorithm, which is quite novel in the genome-editing field. It would strengthen the manuscript if the authors would provide comparisons of its performance with that of other algorithms such as logistic regression, CNN, and RNN as previously shown (Wang et al. Nat. Commun. 2019).
5. The BE-dict web tool does not work.

Minor comments

1. Fig. 1C-F, error bars should be added to the figures. What are the number of target sequences?
2. The correlations between technical replicates are shown in Supplemental Fig. 2. How are technical replicates defined here? Are they independently transfected replicates or independently analyzed (deep sequenced)? What are the correlations between biological replicates?
3. A typo; Page 3, line 86-, ABE_{4max}
4. In the abstract, the authors mention "thousands of lentivirally....". Exact numbers should be given.

Reviewer #2:

Remarks to the Author:

Review "Predicting base editing outcomes with an attention-based deep learning algorithm trained on high-throughput target library screens."

In the presented manuscript entitled "Predicting base editing outcomes with an attention-based deep learning algorithm trained on high-throughput target library screens." the authors present BE-DICT, an algorithm to predict base editing outcomes for four different base editors. Such an algorithm is a timely and relevant deliverable and the authors address it with an overall convincing systematic approach.

Unfortunately -and as the authors address- an alternative prediction tool (BE-Hive) was recently

published substantially taking away novelty of the findings presented herein. BE-Hive represents a more comprehensive tool as it incorporates more base editors and predicts all possible base conversions, while BE-DICT is limited to canonical conversions. Nevertheless, prediction accuracy appears to be at least on par or with BE-Hive in independent small datasets.

Major Issues

1)Line 95-97: The authors chose to exclude those sgRNA/target site vectors from their analysis that showed too few reads or base editing of less than 1.5%. In doing so the authors choose to omit more than 50%/70% of data. This choice might generate an inherent bias in the data resulting also in a biased prediction tool. Such choice must thus be critically discussed. This omission could for example explain why the sgRNA activity prediction tool DeepSpCas9 did not result generate a predictive value, another explanation might be that the sgRNAs that went into the experiment were preselected by some activity selector a priori.

Importantly, all base pair changes appear to be analyzed relative to the expected and putatively synthesized target site, not relative to the sequence confirmed target site cloned into the vector. Mutations occur during synthesis, library generation, PCR, and sequencing, the latter being frequently position dependent. While the overall conversion distribution matches expectations for base editing, it is therefore required to show base pairs without editor as control. Also:

- The authors should explain better how the target loci for library design were selected.
- If possible, the authors should provide control reads of their library in absence of base editors for filtering purposes.
- The authors should critically discuss the stringent and possibly biasing cutoff used within the dataset.
- To validate BE-DICT, the authors should then revisit those omitted datapoint to show that their algorithm predicts well on sgRNAs that were not included in the training data.

2)For all practical purposes and in particular in translational applications a base editing prediction tool must fulfil two purposes: It should predict activity (editing efficiency) and accuracy (precision) of the anticipated edit. Thus, a base editing prediction tool must be able to anticipate which target bases within the activity window will be edited and how selective this choice is. In other words, it must predict the off-target within the on-target window. Ideally it would be beneficial if the selectivity of a base editing prediction tool would also anticipate non-canonical edits. Therefore the authors need to provide an assessment of prediction accuracy in addition to activity. In the simplest form, this could be the anticipated edit versus the accumulative alternative (canonical) edits. Accuracy prediction of BE-DICT could be ideally showcased on the disease-associated human loci. This would also highlight the fraction of their obtained data with particular relevance.

3)The authors state that a web application will be made available under <https://be-dict.org>. Unfortunately the reviewer found this page non-existent during the entire review-period. Any further evaluation of this manuscript critically depends on the existence of deliverables advertised. Should this be a shortcoming on my behalf I hereby apologize.

Minor points

1)Line 84-87: The authors state the coverage of the sgRNA library before transduction of Cas9-base editors (1000x). However, this second transduction event will potentially reduce complexity if not performed in adequate scale. The authors should provide information on the number of cells successfully transduced with base editors.

2)Line 93: Genomic DNA was harvested exclusively at a very early time point, namely four days after transduction with base editors. This choice resulted in very low editing rates and very likely affected the subsequent editing outcomes and thus the predictions of their algorithm. Please qualify the data accordingly.

3)The datasets used to validate BE-DICT versus BE-Hive (Fig. 3C, F and Suppl. Fig. 11) are somewhat limited. Please provide prediction benchmarking for both algorithms on the full datasets of BE-Hive and BE-DICT on itself and reciprocally.

4)Supplementary Fig. 4: Linear scaling of editing efficiency and simple correlation to evaluate DeepSpCas9 scores might be inadequate for the purpose. To make the point that sgRNA activity scorers are not predictive for base editing at all it would be required to show that sgRNA activity scorers do not provide added value on top of BE-DICT also when using other benchmarking matrices. Incorporation of alternative scorers would be beneficial.

5)Similar, a merge of BE-Hive and BE-DICT might increase the predictive power and thus deliver added value.

6)Likely due to the urgent submission of this manuscript the writing appears a bit immature in particular towards the end. Please provide more depth in particular in the discussion or in supplemental figure legends.

We thank both reviewers for their careful evaluation of the manuscript. Please find below a point-by-point reply to your concerns, with the references to the corresponding sections/figures in our revised manuscript. Points raised by the reviewers in black, our responses in green:

REVIEWER COMMENTS

Reviewer #1 (Remarks to the Author):

The authors generated large datasets of CBE and ABE outcomes using lentivirally integrated target sequences and then developed BE-DICT, an attention-based deep learning algorithm capable of predicting base editing outcomes. The following issues should be addressed.

Major comments

1. The novelty of this study has been reduced by two recent studies (Arbab et al. Cell 2020; Song et al. Nat. Biotechnol. 2020). Furthermore, unlike the current study, these published studies considered combinations of base conversions. Given that the majority of users of BE-dict would want to know about such combinations, BE-dict should provide this information as well to reach at least a similar level as these previous studies.

We agree with the reviewer, and we therefore extended BE-DICT to also predict the frequency of combinations of base conversions at a given locus (BE-DICT bystander module - Fig. 5).

2. The overall editing efficiencies are fairly low for both ABE_{max} and CBE4_{max} (median overall editing efficiency ABE_{max}= 2.2%, CBE4_{max}= 2.4%). This finding might be attributable to low transfection efficiencies, given that the authors neither checked the transfection efficiencies nor enriched transfected cells. These low editing efficiencies imply that the large dataset used in this study is of poor quality.

We agree with the reviewer, and therefore optimized our experimental approach: We now stably integrated the self-targeting library (via lentivirus) and the base editor (via Tol2 transposase), and used antibiotic selection to select for cells with dual integrations. This approach substantially increased editing efficiencies (see revised manuscript, Suppl. Fig. 2).

3. Fig. 1E, F; these results should be compared with those of the two recent studies (Arbab et al. Cell 2020; Song et al. Nat. Biotechnol. 2020).

We observed highly similar 3-mer motifs to Arbab et al. and Song et al. In all three studies the most favored motif was TA for ABE, and TC for CBE4. We describe this in the revised manuscript (Page 5, lines 71-79).

4. The authors used an attention algorithm, which is quite novel in the genome-editing field. It would strengthen the manuscript if the authors would provide comparisons of its performance with that of other algorithms such as logistic regression, CNN, and RNN as previously shown (Wang et al. Nat. Commun. 2019).

In the revised manuscript, we compare BE-DICT to a deep conditional autoregressive model (Be-Hive; Arbab et al., 2020) and a convolutional neural network (DeepBaseEditor; Song et al., 2020) – these models were already benchmarked against other baseline models. The performance of the three models was in a similar range (Fig. 5e-g and page 9, lines 179-199).

5. The BE-dict web tool does not work.

BE-DICT is now freely accessible via www.be-dict.org. In the current version of the webpage, we integrated the BE-DICT per-base module. For the BE-DICT bystander module we provide the link to the publicly available GitHub code: <https://github.com/uzh-dqbm-cmi/crispr>. In our next update of the webpage, we will also integrate this module.

Minor comments

1. Fig. 1C-F, error bars should be added to the figures. What are the number of target sequences?

We have added the error bars to the figure, and provide the exact number of sequences analyzed (all loci above mean editing) to the figure legend. The numbers are: 8558 for ABE4max, 9534 for CBE4max, 3416 for ABE8e and 10177 for Target-AID.

2. The correlations between technical replicates are shown in Supplemental Fig. 2. How are technical replicates defined here? Are they independently transfected replicates or independently analyzed (deep sequenced)? What are the correlations between biological replicates?

Replicates have been performed in independent experiments. Cells were transfected on different days with base editors. NGS libraries of the independent replicates have been generated in parallel, and were sequenced on the same Illumina machine in a single run. We describe this in more detail in the revised manuscript (Fig. S3, page 32, lines 618-624).

3. A typo; Page 3, line 86-, ABE4max

We have corrected the typo in the revised manuscript.

4. In the abstract, the authors mention “thousands of lentivirally...”. Exact numbers should be given.

We state the exact number in the abstract in the revised manuscript.

Reviewer #2 (Remarks to the Author):

Review “Predicting base editing outcomes with an attention-based deep learning algorithm trained on high-throughput target library screens.”

In the presented manuscript entitled “Predicting base editing outcomes with an attention-based deep learning algorithm trained on high-throughput target library screens.” the authors present BE-DICT, an algorithm to predict base editing outcomes for four different base editors. Such an algorithm is a timely and relevant deliverable and the authors address it with an overall convincing systematic approach.

Unfortunately -and as the authors address- an alternative prediction tool (BE-Hive) was recently published substantially taking away novelty of the findings presented herein. BE-Hive represents a more comprehensive tool as it incorporates more base editors and predicts all possible base conversions, while BE-DICT is limited to canonical conversions. Nevertheless, prediction accuracy appears to be at least on par or with BE-Hive in independent small datasets.

We thank the reviewer for acknowledging the value of our tool. In the revised manuscript we also implemented a BE-DICT module for predicting bystander edits within the protospacer sequence (Please see response to point 2 for details).

Major Issues

1)Line 95-97: The authors chose to exclude those sgRNA/target site vectors from their analysis that showed too few reads or base editing of less than 1.5%. In doing so the authors choose to omit more than 50%/70% of data. This choice might generate an inherent bias in the data resulting also in a biased prediction tool. Such choice must thus be critically discussed. This omission could for example explain why the sgRNA activity prediction tool DeepSpCas9 did not result generate a predictive value, another explanation might be that the sgRNAs that went into the experiment were preselected by some activity selector a priori.

We have adapted our experimental protocol to increase the overall editing rates (see response to rev. 1, comment 2). Indeed, generating a bias in the machine learning tool by pre-selecting only edited sequences is a concern. Therefore, we generated bins of edited and unedited sequences, and added sequences of both categories to train and test the model. We describe this procedure in the revised manuscript (page 6, lines 103-108).

Importantly, all base pair changes appear to be analyzed relative to the expected and putatively synthesized target site, not relative to the sequence confirmed target site cloned into the vector. Mutations occur during synthesis, library generation, PCR, and sequencing, the latter being frequently position dependent. While the overall conversion distribution matches expectations for base editing, it is therefore required to show base pairs without editor as control.

We have now sequenced untreated library cells as a control (see Supplementary Data File S1). The background error rate at the target sequences was consistently below 1%.

Also:

- The authors should explain better how the target loci for library design were selected.

In the revised manuscript we explain the library design in detail in the Materials and Methods section (Oligonucleotide-library design; page 12, lines 231-248).

- If possible, the authors should provide control reads of their library in absence of base editors for filtering purposes.

We include the control reads in the Supplementary Data File S1.

- The authors should critically discuss the stringent and possibly biasing cutoff used within the dataset.

Our machine learning model uses a binary input, where we defined base with editing rates above or equal to the mean as edited, and below the mean as unedited. We believe that this chosen cut-off was validated, as the models trained on these sequences were competitive when tested on external datasets (Figure 3 and 5E,F, Suppl. Fig. 7,8).

- To validate BE-DICT, the authors should then revisit those omitted datapoint to show that their algorithm predicts well on sgRNAs that were not included in the training data.

The number of edited sequences (in which at least one base had an editing rate above or equal to the mean as stated above) was lower than that of the unedited sequences. We therefore used all edited sequences and a sample of the unedited sequences (in a ratio of 4:1) for training and testing of our models. The omission of part of the unedited sequences for training did not influence the generalizability of our approach, as in particular shown by the performance of the model on external datasets (Figure 3 and 5E,F, Suppl. Fig. 7,8).

2) For all practical purposes and in particular in translational applications a base editing prediction tool must fulfil two purposes: It should predict activity (editing efficiency) and accuracy (precision) of the anticipated edit. Thus, a base editing prediction tool must be able to anticipate which target bases within the activity window will be edited and how selective this choice is. In other words, it must predict the off-target within the on-target window. Ideally it would be beneficial if the selectivity of a base editing prediction tool would also anticipate non-canonical edits. Therefore, the authors need to provide an assessment of prediction accuracy in addition to activity. In the simplest form, this could be the anticipated edit versus the accumulative alternative (canonical) edits. Accuracy prediction of BE-DICT could be ideally showcased on the disease-associated human loci. This would also highlight the fraction of their obtained data with particular relevance.

We agree that the possibility of predicting bystander base editing in the target sequence would be a great benefit. We have therefore extended our model to predict combinations of base conversions (accumulative alternative edits) in the target sequences (BE-DICT bystander module - Fig. 5).

3) The authors state that a web application will be made available under <https://be-dict.org>. Unfortunately the reviewer found this page non-existent during the entire review-period. Any further evaluation of this manuscript critically depends on the existence of deliverables advertised. Should this be a shortcoming on my behalf I hereby apologize.

BE-DICT is now freely accessible via www.be-dict.org. In the current version of the webpage, we integrated the BE-DICT per-base module. For the BE-DICT haplotype module we provide the link to the publicly available GitHub code: <https://github.com/uzh-dqbm-cmi/crispr>. In our next update of the webpage, we will also integrate this module.

Minor points

1) Line 84-87: The authors state the coverage of the sgRNA library before transduction of Cas9-base editors (1000x). However, this second transduction event will potentially reduce complexity if not performed in adequate scale. The authors should provide information on the number of cells successfully transduced with base editors.

During the revision we have optimized our experimental strategy. We re-screened the target sequences using stable integration of the self-targeting library (via lentivirus) and the base editor (via Tol2 transposase). We used antibiotic selection to select for cells that express the base editor and that contain an integrated self-targeting sgRNAs. We transfected 70 million cells library-containing cells per replicate with the base editor, of which approx. 50% died during selection. We extracted genomic DNA from sufficient cells to achieve 1600 x coverage per replicate. We describe the experimental procedure in more detail in the revised manuscript (Page 15, lines 320-323).

2) Line 93: Genomic DNA was harvested exclusively at a very early time point, namely four days after transduction with base editors. This choice resulted in very low editing rates and very likely affected the subsequent editing outcomes and thus the predictions of their algorithm. Please qualify the data accordingly.

In the revised workflow we performed antibiotic selection and extended the time point of harvesting the genomic DNA to 10 days post transfection with the base editors. This significantly increased editing rates.

3)The datasets used to validate BE-DICT versus BE-Hive (Fig. 3C, F and Suppl. Fig. 11) are somewhat limited. Please provide prediction benchmarking for both algorithms on the full datasets of BE-Hive and BE-DICT on itself and reciprocally.

To compare performance between BE-Hive and BE_DICT, we predicted ABE editing outcomes on a 'third-party' dataset of 438 edited sites published by Song et al. (Song et al., NatBiot 2020). We observed comparable performance of both models on this independent dataset (BE-DICT $r=0.91$; BE-Hive $r=0.87$) (Fig. 5e, g).

4)Supplementary Fig. 4: Linear scaling of editing efficiency and simple correlation to evaluate DeepSpCas9 scores might be inadequate for the purpose. To make the point that sgRNA activity scorer are not predictive for base editing at all it would be required to show that sgRNA activity scorer do not provide added value on top of BE-DICT also when using other benchmarking matrices. Incorporation of alternative scorers would be beneficial.

We agree that the comparison performed in the initial manuscript might not be appropriate. We therefore remove the evaluation of DeepSpCas9 for base editing prediction.

5)Similar, a merge of BE-Hive and BE-DICT might increase the predictive power and thus deliver added value.

We have extended our per-base model to predict edit combination probabilities, like BE-Hive and DeepBaseEditor. This allowed us to directly compare performance of the three models, and to show that BE-DICT is on par with BE-Hive and DeepBaseEditor. However, due to the different architecture, it is not feasible to merge the models. Since all three models are publicly available, researchers could use them in parallel to predict editing at the desired locus.

6)Likely due to the urgent submission of this manuscript the writing appears a bit immature in particular towards the end. Please provide more depth in particular in the discussion or in supplemental figure legends.

We extended the discussion in the revised manuscript.

Reviewers' Comments:

Reviewer #1:

Remarks to the Author:

I thank the authors for significantly improving the manuscript by performing additional experiments and analyses, which successfully addressed most of my concerns, except for one major point. I think that the major point can be addressed without additional experiments. In addition, there are three very minor points that can be easily addressed without additional experiments.

Major comment

1. The authors have now compared their models with BE-Hive ABE and ABE_proportion. This comparison improved the manuscript. However, the authors have not compared their models with BE-Hive CBE and CBE_proportion. More importantly, the authors did not compare their efficiency-predicting model with BE-Hive, ABE_efficiency, and CBE_efficiency. Without extensive comparisons, it is difficult to determine whether the computational models developed by the authors reach at least the level of these two published models. Given that many data sets of base editor efficiencies and outcomes are already available from the previous two publications (Arbab et al 2020; Song et al 2020) as well as those from the authors, it is now much easier to compare the three models (authors' model, BE-Hive, DeepBaseEditor). Even if the authors' model does not turn out to be the best, I would still recommend the publication of this interesting manuscript as long as the authors extensively compare the three models using diverse data sets regarding both base editing efficiency and outcomes because these comparisons will provide a guide to readers in choosing appropriate computational models. To summarize the extensive comparison results, the authors may use heatmaps of Pearson correlation coefficients and/or Spearman correlation coefficients as previously done (Figure 4 of Haeussler et al. Genome Biol 17, 148 (2016)., <https://doi.org/10.1186/s13059-016-1012-2>).

Minor comments

1. Fig. 5D, the maximum value for correlation (r) should be "1" rather than "100".
2. line 622, "Spearman's correlation coefficient (r^2) is shown.". Shouldn't it be "Pearson's ..."?
3. Integration of the BE-DICT bystander module into the web tool should be checked. The authors say that they will update this part later but this point should be confirmed before this manuscript is accepted.

Reviewer #2:

Remarks to the Author:

The revised manuscript presents the BE-DICT algorithm as an sgRNA selection tool reaching comparable predictive value as other such tools.

The authors have implemented the suggestions by the reviewers into the revised manuscript, text and figures are mature and solid. I therefore support publication.

We again thank both reviewers for evaluating the manuscript. Please find below a point-by-point reply to your concerns, with the references to the corresponding sections/figures in our revised manuscript (our response highlighted in green).

REVIEWER COMMENTS

Reviewer #1 (Remarks to the Author):

I thank the authors for significantly improving the manuscript by performing additional experiments and analyses, which successfully addressed most of my concerns, except for one major point. I think that the major point can be addressed without additional experiments. In addition, there are three very minor points that can be easily addressed without additional experiments.

Major comment

1. The authors have now compared their models with BE-Hive ABE and ABE_proportion. This comparison improved the manuscript. However, the authors have not compared their models with BE-Hive CBE and CBE_proportion. More importantly, the authors did not compare their efficiency-predicting model with BE-Hive, ABE_efficiency, and CBE_efficiency. Without extensive comparisons, it is difficult to determine whether the computational models developed by the authors reach at least the level of these two published models. Given that many data sets of base editor efficiencies and outcomes are already available from the previous two publications (Arbab et al 2020; Song et al 2020) as well as those from the authors, it is now much easier to compare the three models (authors' model, BE-Hive, DeepBaseEditor). Even if the authors' model does not turn out to be the best, I would still recommend the publication of this interesting manuscript as long as the authors extensively compare the three models using diverse data sets regarding both base editing efficiency and outcomes because these comparisons will provide a guide to readers in choosing appropriate computational models. To summarize the extensive comparison results, the authors may use heatmaps of Pearson correlation coefficients and/or Spearman correlation coefficients as previously done (Figure 4 of Haeussler et al. Genome Biol 17, 148 (2016)., <https://doi.org/10.1186/s13059-016-1012-2>).

We thank the reviewer for the assessment of our revised manuscript. Following the reviewer's suggestion, we have now performed an extensive comparison of BE-HIVE (Arbab et al., 2020), DeepBaseEditor (Song et al., 2020) and BE-DICT (this study) using ABE and CBE datasets from each study. In summary, we compared the editing outcomes and editing efficiency of >3300 unique sequences for ABE and >3500 for CBE (Fig. 5d-h), and plot the Pearson correlation (Fig. 5e-h) and Spearman correlation (Fig. S10) in heatmaps as suggested.

For **editing outcomes**, we assess the performance in predicting the *proportion* of edited alleles (i.e. sequences with a unique combination of target base and bystander mutations). To this end, we calculated and compared the proportion of a particular *editing combination* over *all editing combinations* at the >3300 loci (ABE) and >3500 loci (CBE) across the different models. The results are shown Figure 5e,f (“proportions of edited outcomes”).

For **editing efficiency**, we assess and compare the performance of each model in predicting the *proportion* of the *editing combinations* over *all outcomes*, including the unedited wild-type sequence (Fig. 5g, h, “proportion of all outcomes”).

Notably, the BE-DICT bystander model can directly predict the proportion of all outcomes (edited and non-edited outcomes), whereas for BE-HIVE and DeepBaseEditor models, these values are calculated by multiplying the proportions of edited outcomes with the overall locus editing

efficiency. As this difference between the BE-DICT bystander model and the other published models was not well described in the previous version of our manuscript, we provide a more detailed explanation in the revised version (Line 197-203).

Also, we believe there has been a misunderstanding about the relationship between the BE-DICT per-base module and bystander module. Unlike in the other two publications, we did not create an efficiency model and an outcome model. The BE-DICT bystander module is an extension of the BE-DICT per-base model (which predicts editing of single bases), and directly predicts editing outcomes and efficiencies at a given locus. Thus, we could directly benchmark the BE-DICT bystander module to the published efficiency- and outcome models, and do not see a large benefit in also benchmarking the BE-DICT per-base module – which would also be difficult, as the BE-DICT per-base module predicts the probability of editing of a respective base and not of a locus.

Minor comments

1. Fig. 5D, the maximum value for correlation (r) should be “1” rather than “100”.

We have corrected the error in the revised manuscript.

2. line 622, “Spearman’s correlation coefficient (r^2) is shown.”. Shouldn’t it be “Pearson’s ...”?

We have corrected the error in the revised manuscript.

3. Integration of the BE-DICT bystander module into the web tool should be checked. The authors say that they will update this part later but this point should be confirmed before this manuscript is accepted.

We have integrated the BE-DICT bystander module into the web tool (www.be-dict.org).

Reviewer #2 (Remarks to the Author):

The revised manuscript presents the BE-DICT algorithm as an sgRNA selection tool reaching comparable predictive value as other such tools.

The authors have implemented the suggestions by the reviewers into the revised manuscript, text and figures are mature and solid. I therefore support publication.

We thank the reviewer for the evaluation and for supporting publication of the manuscript.

Reviewers' Comments:

Reviewer #1:

Remarks to the Author:

I appreciate that the authors did additional analyses to address my concerns.

Major comment

1. The authors have now added comparisons of the three models as we suggested. These comparisons would be a big selling point for the manuscript. However, we found that some of the correlation coefficients, which were obtained using data sets from previous publications as test data sets, shown in the newly added figures are drastically different from those published in the relevant previous publications. The authors should describe the reason for these incompatibilities. To help the authors find the reason for those incompatibilities, we have checked some of the raw data in the manuscript. We found that there are some errors in the authors' analyses. Seq_0 shown in the sheet "DEEPBE_own_prop_CBE" of the "266000_2_data_set_5626128_ql6br9" file has TGG as the PAM sequence. However, it appears that the authors analyzed the PAM sequence as GGG. A similar error was observed in another sheet (Seq_1 of DEEPBE_own_prop_ABE). These errors suggest that the newly added results could be wrong. As stated in the previous round of review, I think that fair comparisons between the three groups of computational models would make this manuscript attractive even if BE-DICT does not turn out to be the best model. However, the current inaccurate analyses kill the biggest selling point of the manuscript.

Minor comments

1. It is easier to interpret heatmaps when the test datasets and the relevant models are in the same order. Because the test datasets are ordered Arbab et al., Song et al., and this study (from the left), the models should be ordered BE-Hive, DeepABE or DeepCBE, and BE-DICT (from the bottom).
2. We believe that target sequences were not shared between training and test datasets in all analyses of the current manuscript. However, after noticing the errors described in major comment 1, we recommend checking this point again to make the analyses accurate and fair.
3. After noticing the incompatibilities described in major comment 1, we would like to suggest providing a correlation dot graph for each correlation coefficient shown in the newly added Fig. 5e-h as Supplementary Figures as shown in Supplementary Fig. 8. We think that the total number of graphs will be $9 \times 4 = 36$.

We again thank the reviewer for evaluating our manuscript. Please find below a point-by-point reply to your concerns, with the references to the corresponding sections/figures in our revised manuscript (responses highlighted in green).

REVIEWER COMMENTS

Reviewer #1 (Remarks to the Author):

I appreciate that the authors did additional analyses to address my concerns.

Major comment

1. The authors have now added comparisons of the three models as we suggested. These comparisons would be a big selling point for the manuscript. However, we found that some of the correlation coefficients, which were obtained using data sets from previous publications as test data sets, shown in the newly added figures are drastically different from those published in the relevant previous publications. The authors should describe the reason for these incompatibilities. To help the authors find the reason for those incompatibilities, we have checked some of the raw data in the manuscript. We found that there are some errors in the authors' analyses.

Seq_0 shown in the sheet "DEEPBE_own_prop_CBE" of the "266000_2_data_set_5626128_ql6br9" file has TGG as the PAM sequence. However, it appears that the authors analyzed the PAM sequence as GGG. A similar error was observed in another sheet (Seq_1 of DEEPBE_own_prop_ABE).

These errors suggest that the newly added results could be wrong. As stated in the previous round of review, I think that fair comparisons between the three groups of computational models would make this manuscript attractive even if BE-DICT does not turn out to be the best model. However, the current inaccurate analyses kill the biggest selling point of the manuscript.

We thank the reviewer for spotting this error in our analysis. Indeed, in some target sequences the wrong PAM site was selected. At these sites the protospacer sequence between predicted and experimentally observed editing was not identical. In the revised manuscript this error was corrected, and always the exact same protospacer sequences that were experimentally targeted were predicted.

Correcting this error significantly improved the R values in our analysis, and importantly our R values for DeepBE (predicted proportions to experimentally observed proportions) are now almost identical to the reported values in Song et al. ($R= 0.95$ for CBE and 0.96 for ABE). Of note, the R values of our plots 'proportions of all outcomes' cannot be compared with the R values of Song et al. 'predicted frequency' (Fig. 2b, c – top right panel), as they did not include the unedited outcome in their analysis.

Minor comments

1. It is easier to interpret heatmaps when the test datasets and the relevant models are in the same order. Because the test datasets are ordered Arbab et al., Song et al., and this study (from the left), the models should be ordered BE-Hive, DeepABE or DeepCBE, and BE-DICT (from the bottom).

We re-arranged the order of plots as suggested to facilitate heatmap interpretation. We also keep the same arrangement for the newly added dot plots in Figure S10.

2. We believe that target sequences were not shared between training and test datasets in all analyses of the current manuscript. However, after noticing the errors described in major comment 1, we recommend checking this point again to make the analyses accurate and fair.

For our datasets we explicitly state in the Suppl. Table 1 (Data File S1) which sequences were used for training and testing. When we compared the models, we only used the test dataset. Song et al. also stated which sequences were used for training and testing, and we again only used their test dataset for comparison. Arbab et al. did not state which sequences were used for training and testing, and we therefore took a random set (20% of the data set) of 1265 sequences for ABE and 1358 sequences for CBE when comparing the three models. Importantly, all sequences used for comparison of the 3 models are shown in Data File S3.

3. After noticing the incompatibilities described in major comment 1, we would like to suggest providing a correlation dot graph for each correlation coefficient shown in the newly added Fig. 5e-h as Supplementary Figures as shown in Supplementary Fig. 8. We think that the total number of graphs will be $9 \times 4 = 36$.

As suggested, we added the dot plots for all 36 correlations shown in Fig. 5e-h in Figure S10.

Reviewers' Comments:

Reviewer #1:

Remarks to the Author:

The authors have successfully addressed my concerns. This interesting and informative manuscript is now complete enough for publication.